# A Rapid Crosslinkable Maleimide-Modified Hyaluronic Acid and Gelatin Hydrogel Delivery System for Regenerative Applications

**DOI:** 10.3390/gels7010013

**Published:** 2021-02-01

**Authors:** Kyung Min Yoo, Sean V. Murphy, Aleksander Skardal

**Affiliations:** 1Wake Forest Institute for Regenerative Medicine, Wake Forest School of Medicine, 391 Technology Way, Winston-Salem, NC 27101, USA; kmyoo@wakehealth.edu; 2Department of Biomedical Engineering, The Ohio State University, Fontana Labs., 140 W. 19th Ave, Columbus, OH 43210, USA; 3Ohio State University and Arthur G. James Comprehensive Cancer Center, Columbus, OH 43210, USA

**Keywords:** hydrogel, hyaluronic acid, gelatin, rapid crosslinking, maleimide

## Abstract

Hydrogels have played a significant role in many applications of regenerative medicine and tissue engineering due to their versatile properties in realizing design and functional requirements. However, as bioengineered solutions are translated towards clinical application, new hurdles and subsequent material requirements can arise. For example, in applications such as cell encapsulation, drug delivery, and biofabrication, in a clinical setting, hydrogels benefit from being comprised of natural extracellular matrix-based materials, but with defined, controllable, and modular properties. Advantages for these clinical applications include ultraviolet light-free and rapid polymerization crosslinking kinetics, and a cell-friendly crosslinking environment that supports cell encapsulation or in situ crosslinking in the presence of cells and tissue. Here we describe the synthesis and characterization of maleimide-modified hyaluronic acid (HA) and gelatin, which are crosslinked using a bifunctional thiolated polyethylene glycol (PEG) crosslinker. Synthesized products were evaluated by proton nuclear magnetic resonance (NMR), ultraviolet visibility spectrometry, size exclusion chromatography, and pH sensitivity, which confirmed successful HA and gelatin modification, molecular weights, and readiness for crosslinking. Gelation testing both by visual and NMR confirmed successful and rapid crosslinking, after which the hydrogels were characterized by rheology, swelling assays, protein release, and barrier function against dextran diffusion. Lastly, biocompatibility was assessed in the presence of human dermal fibroblasts and keratinocytes, showing continued proliferation with or without the hydrogel. These initial studies present a defined, and well-characterized extracellular matrix (ECM)-based hydrogel platform with versatile properties suitable for a variety of applications in regenerative medicine and tissue engineering.

## 1. Introduction

Hydrogel biomaterials have shown immense potential in a variety of regenerative medicine and tissue engineering applications [1,2,3]. These applications range from delivery vehicles for drugs and cell therapies, to serving as a bioengineered environment for creating 3D tissue constructs via bioprinting and other biofabrication approaches, to cell-free biomaterial therapies [4,5,6,7,8]. Across these broad applications, a wide variety of hydrogel types have been explored and implemented. These include synthetic hydrogels such as polyethylene glycol (PEG) and its derivative PEG diacrylate (PEGDA) [9,10,11,12,13], and natural hydrogels such as collagen type I, fibrin, glycosaminoglycans such as hyaluronic acid (HA) [14,15,16,17,18,19], and less defined matrices such as Matrigel [20,21].

In the context of either cell therapy or biomaterial-only therapeutic deposition, a limiting step in successful utility of a biomaterial is how one can control the sol-gel transition (generally through a crosslinking reaction), a dynamic process in which a liquid or less viscous hydrogel is deposited to a particular site, followed by a transition to a more solid and robust material state in order for the hydrogel to hold its shape or geometry. For example, in the context of applications such as 3D bioprinting or in situ delivery to a wound, if one deposits a material too early (prior to crosslinking), one may end up depositing a fluid, resulting in essentially a flat puddle of material. Conversely, if one deposits a material too late, it may crosslink early and end up clogging the printer or syringe delivery mechanism [22,23]. In wound healing specifically, if one is trying to administer a hydrogel—with or without cells—as a wound healing therapy, one runs a risk of improper wound coverage of a therapeutic material if the sol-gel transition fails and results in hydrogel precursor liquid falling or sluffing off of an irregularly shaped or not flat wound site [24,25]. As such, direct on-site polymerization of a hydrogel product is a crucial feature that has widely been overlooked in developing wound healing therapies. To be specific, one cannot simply apply some hydrogels commonly used in cell culture, such as rat tail collagen or Matrigel materials to a wound and wait 20–60 min for them to crosslink in place. Rather, a fast-crosslinking solution is significantly more amenable in a clinical setting.

As described above, hydrogels come in both synthetic and natural varieties. While synthetic biomaterials have shown utility in a variety of applications, both as hydrogels such as PEG-based materials [9,10,11,12,13], and as load-bearing orthopedic implants generated using synthetic polymers such as poly(lactic-co-glycolic acid) (PLGA) and polycaprolactone (PCL) [26,27,28], our team has largely focused on the use of naturally derived hydrogels. In general, natural hydrogels include those formed from collagen, gelatin, fibrin, HA, and other individual ECM components. On the other end of the spectrum are more complex natural hydrogels such as Matrigel and those derived from whole decellularized tissues. These complex ECM hydrogels have immense potency for supporting cell types of many lineages, including difficult to culture cell populations, due to preservation of ECM-bound growth factors and cytokines [29,30,31,32]. However, these come with a significant drawback in that they are essentially uncharacterized black boxes in which their complete compositions cannot be completely defined or controlled, posing hurdles in terms of regulatory hurdles and clinical translation [22,33,34]. While we have used undefined ECM materials in our hydrogels [31,35,36,37], we now focus on building ECM-derived hydrogels from the ground up, therefore ensuring that the entire system’s composition is well-defined. These hydrogel platforms are considerably simpler and more defined than Matrigel, while offering capabilities of tissue-specific ECM complexity. They are generally modified with functional chemical groups to aid with crosslinking and to imbue other attachment points, thus enabling manipulation of biochemical complexity by modular inclusion synthetically modified natural proteins, peptides, and polymers.

To address these challenges, we aimed to combine defined ECM-derived components, rapid gelation kinetics for controlled deposition, crosslinking conditions amenable to high cell viability that do not employ photocrosslinking, to generate a natural ECM-based hydrogel platform with utility across applications in regenerative medicine and tissue engineering. We employ HA and gelatin base materials, based on our long-standing expertise in deploying these natural materials within a variety of hydrogel systems and subsequent applications including wound healing [25,38], bioprinting [17,39,40,41,42,43], and organoid/tissue chip platforms [15,36,44,45,46,47]. These past studies have largely utilized thiol, acrylate, or methacrylate modified HA, gelatin, and collagen, which have been effective tools. However, the chemistries involved have relied on ultraviolet (UV) photopolymerization to support rapid crosslinking. While these rapid and easy to control gelation kinetics are beneficial, the UV irradiation could be undesirable in the clinic due to potential toxic effects of UV exposure to cells. Without UV photopolymerization, the gelation kinetics of these previous hydrogel systems are relatively slow. As a result, herein we describe the synthesis of a hydrogel system comprised of maleimide-modified HA and gelatin to support UV-free rapid crosslinking with a thiol-functionalized crosslinker, complete with chemical characterization of synthesized components, hydrogel material testing, and in vitro biocompatibility assessment. The studies described focus on the development and testing of this hydrogel system, with a plan for future deployment in preclinical tissue regeneration applications.

## 2. Results and Discussion

### 2.1. HA-Mal and Gel-Mal Syntheses and Evaluation

HA-maleimide (HA-Mal) and gelatin-maleimide (Gel-Mal) were successfully synthesized via one-pot reactions (Figure 1). Unlike other published methods, which involve performing a two-step protocol and organic solvents [48,49], this maleimide functionalization method occurred under aqueous conditions using water soluble reagents, 1-Ethyl-3-(3-dimethylaminopropyl) carbodiimide) (EDC) and *N*-hydroxysuccinimide (NHS), to activate the carboxylic acid groups on hyaluronic acid (HA) and gelatin for 30 min prior to adding the 1-(2-Aminoethyl) maleimide molecule that forms the amide linkage. Following the overnight reactions, crude mixture solutions were purified by dialysis under acidic conditions to remove excess unreacted reagents and byproducts, frozen and lyophilized. The resulting HA-Mal and Gel-Mal materials were characterized by nuclear magnetic resonance (NMR) spectroscopy to confirm the modification as well as the purity (Figure 2a). The singlet peak at 6.8 ppm corresponding to the two symmetrical protons on the maleimide ring was present both HA-Mal and Gel-Mal spectra. This confirms that both HA and gelatin were successfully covalently modified with maleimide groups. For HA-Mal specifically, the degree of substitution (DS%) was determined to be 7.2 ± 0.5% (n = 7) by normalizing the integration of the maleimide peak to the peak at 1.9 ppm that is associated with the three protons on the *N*-acetyl group on the *N*-acetyl-glucosamine moiety in HA. The NMR spectra were also analyzed for the purity in both materials, which was confirmed by the absence of additional peaks associated with residual reagents and byproducts. Performing dialysis of the crude mixture solutions in acidic water (pH < 5) for 3 days was necessary to remove the impurities entirely.

Maintaining the pH of the solutions was found to be critical throughout the reaction process due to the sensitivity of the maleimide group under neutral and basic conditions. To further investigate the stability of the maleimide group under various pH conditions, UV–Vis spectroscopy was used to monitor the change in absorbance values at 295 nm of maleimide dissolved in solutions of various pH (3–8) for 4 days (Appendix A). It was shown that pH had a significant impact in the conservation of the maleimide group. Over a period of 96 h, the maleimide dissolved in solutions pH < 5 was conserved by sustaining absorbances of 0.90–0.95. However, absorbance values began to decrease within 24 h when dissolved in solutions greater than pH 6. As pH increases, the molecule loses its conjugation system and thus reduces the absorbance values, which continues over time. Therefore, it was concluded that the maleimide group is pH sensitive and that it should be below pH 5 throughout the reaction and purification processes in order to conserve the maleimide for hydrogel crosslinking. 

UV–Vis spectroscopy was also used to characterize and quantify the maleimide concentration of the resulting HA-Mal and Gel-Mal materials. While the max absorbance value of the maleimide was already determined to be measured at a 295 nm, HA-Mal and Gel-Mal solutions were scanned between 200–400 nm to confirm the maximum absorbance wavelength (λ_max_). As shown in Figure 2b, both the unmodified HA (red) and gelatin (blue) did not result in a maximum absorbance peak above 280 nm, while both HA-Mal (dotted red) and Gel-Mal (dotted blue) materials expressed a prominent maximum absorbance peak at 300 nm. While it was found at a wavelength slightly higher than the small molecule maleimide, this result offers confirmation that HA and gelatin were modified with maleimides. Once the λ_max_ was set to 300 nm, the maleimide concentration (µmol/mg) was determined using a standard curve of 1-(2-Aminoethyl)maleimide hydrochloride and absorbance values of HA-Mal and Gel-Mal with absorbance measurements of HA and gelatin subtracted from them. The resulting concentrations of maleimide in HA-Mal and Gel-Mal were 0.34 ± 0.06 µmol/mg (*n* = 4) and 0.11 ± 0.03 µmol/mg (*n* = 4), respectively. In comparison to HA-Mal, the maleimide concentration in Gel-Mal was not as high because the number of carboxylic acid (COOH) groups in gelatin (found in glutamic acid and aspartic acid amino acid residues) is known to be less frequent than the COOH groups on the repeating glucuronic acid units of HA. 

Lastly, the molecular weight of HA-Mal was quantified using SEC-MALS (size-exclusion chromatography with multi-angle light scattering). Initially, there was concern of the potential acidic hydrolysis of HA that would decrease the average molecular weight during synthesis and purification. However, instead of seeing a suspected decrease in molecular weight of HA-Mal compared to the HA, the average molecular weights of the peaks with elution times of 10–14 min increased (Figure 2c). The HA that was purchased from Lifecore was characterized to have an average molecular weight of 200 kDa, but instead it was determined to be 142 ± 4.4 kDa. Compared to this, HA-Mal from 4 different synthesis batches resulted in molecular weights of 203 ± 2.6, 271 ± 4.5, 200 ± 3.48, and 174 ± 1.76 kDa, demonstrating that despite batch-to-batch variability, the addition of the maleimide groups consistently increased the overall molecular weight of the HA component. It should be noted that Gel-Mal was not put through the HPLC system due to the nature of gelatin varying considerably in terms of molecular weight. As stated, SEC-MALS revealed some variation in molecular weight of HA-Mal batches. However, this is of relatively minor concern as upon crosslinking into a hydrogel, the HA-Mal polysaccharide chains join into a single macromolecular network, where the original polysaccharide molecular weight is less of a driving factor of hydrogel material properties and more derived from the crosslinking density, which is based on the relative molecular weights of polysaccharide regions between crosslinking points. 

### 2.2. Hydrogel Preparation

Crosslinked hydrogels were prepared using 3 components, HA-Mal, Gel-Mal and a PEGDSH crosslinker (Figure 3a), which form a macromolecular polysaccharide and protein network, visualized in Figure 3b,c. In this particular biomaterial, the crosslinking mechanism uses the Michael-Addition reaction between the thiolated groups on the PEG and the maleimide groups on the HA-Mal and Gel-Mal. After having synthesized HA-Mal using batches of HA with a range of average molecular weights (60–200 kDa), and Gel-Mal with gelatin with different bloom strengths (90–100, 175, 300 g), and then combining them at varying ratios with thiolated PEG crosslinkers of several weights (3.4 kDa, 10 kDa) and arms (2, 4), two formulations (HMGM 1 and HMGM 2) were developed that resulted in self-standing, easily controllable hydrogels (Appendix A). Appendix A describes the variety of formulations tested wih qualitative assessment of whether or not they were self standing and could be easily extrudable for subsequent applications. It should be noted that our team has used 3.4 k PEG crosslinkers (PEG diacrylate) in a wide variety of biomaterial, tissue engineering, and in vivo studies [25,41,44,47,50,51,52]. When paired with natural polysaccharides of much larger molecular weight, these mixtures formed soft, pliable, but robust hydrogels that weer employed in wound healing studies and studies focused on generating 3D organoids for drug screening and disease modeling. As such, we expected that the 3.4 kDa would be optimal, and it was the most effective. HMGM 1 consisted of final concentrations of HA-Mal, Gel-Mal and PEGDSH of 1%, 0.4%, 0.25% *w*/*v*, respectively, and HMGM 2 had concentrations of 0.5, 0.4, 0.25% *w*/*v*. Materials had a resulting pH of 3.0 (to ensure the maleimides were conserved) and were dissolved in 10× phosphate-buffered solution (PBS), which increased the individual component solutions to a pH 6. When the solutions were combined and mixed, crosslinking occurred and had a final pH of 6.5–7. It was observed that solutions did not successfully crosslink (remained a liquid) when the overall pH was <6. Fortunately, these final conditions upon mixing fell near neutral pH, meaning that the crosslinking reaction was likely safe in the presence of cells, thus enabling the use of the hydrogel for cell-based applications such as cell encapsulation or in situ crosslinking.

### 2.3. Hydrogel Gelation Testing

To confirm proper hydrogel gelation, we used an initial visual observation, followed by NMR evaluation of the thiol-maleimide reaction. Hydrogels were formed as described below, using HA-Mal, Gel-Mal and PEGDSH components, but inside glass vials for hydrogel inversion tests. Figure 4a(i) demonstrates the inability for the materials to form a gel due to the absence of the PEGDSH crosslinker, with the uncrosslinked solution falling downwards in the vial. In Figure 4a(ii),(iii), visual images correspond to hydrogels labeled as HMGM 1 and HMGM 2, which used 2 different molecular weights and bloom strengths of HA and gelatin. The addition of PEGDSH in both formulations caused gelation to occur within 5 s resulting in a self-standing gel, that holds its location within the vial.

NMR spectroscopy was then used to confirm the chemical crosslinking reaction between the maleimide and thiol groups. As shown in Figure 4b, highlighted regions of interest offer insight into the state of the maleimide groups by observing the presence, reduction or disappearance of the corresponding peak at 6.8 ppm. It was observed that components without the PEG crosslinker had visual singlet peaks at 6.8 ppm. However, when the PEGDSH crosslinker was added at the appropriate concentrations, it reacted with all the maleimides in HA-Mal, most of all maleimides in Gel-Mal, and the most of all maleimides in the HA-Mal+Gel-Mal mix, based on the the singlet peaks at 6.8 ppm having disappeared or were significantly reduced. This indicates the successful reactions between the thiol and maleimide groups. 

### 2.4. Hydrogel Physical Characterization

Hydrogels then underwent a series of material characterization assays, including rheological assessment, swelling assays, a mechanical barrier to diffusion assay, and total protein release quantification. For rheological testing, each hydrogel sample used to determine the mechanical properties was prepared right before each test. Oscillation strain sweeps from 0.1–100% strain at 1 Hz frequency indicated a hydrogel that is self-standing, but with relatively weak mechanical properties; in other words, soft (Figure 5a). It was determined that the average storage modulus value averages (G’) averaged 350 Pa between 0.1–1% oscillation strain, and then around 300 Pa after 1%. While this hydrogel is not very stiff and rigid, seeing that the material is able to sustain the G’ values between 2 ranges of strain indicates the ability to be flexible, but still hold its shape. Furthermore, G’ values between 300 and 350 Pa are similar to those of several hydrogel formulations our team has successfully employed in past studies. Of particular note is the commercially available thiolated HA-based hydrogel kit (Hystem) which our team has used in its commercial form and in customized formulations and variations in applications across tissue engineering and regeneration, including wound healing [24,25], organoid biofabrication [47,53], bioprinting [23,41,42,43,54], and organ-on-a-chip platforms [15,25,36,44,50]. As such, despite being relatively soft, the HA-Mal and Gel-Mal-based hydrogel system will likely be sufficiently robust for similar uses. Should stiffer materials be desired, we can employ past strategies such as further modulation of concentrations, reduction in crosslinker molecular weight, or use of multi-arm crosslinkers to increase crosslinking density [25,41,43,55].

To measure swelling, hydrogels were prepared as described below and weighed upon gelation (initial mass) and then over a course of 24 h (swelling masses). The swelling ratio was defined as the ratio of the difference in mass between the swollen and initial hydrogel to the mass of the initial hydrogel. Swelling properties were observed in both hydrogels HMGM 1 and 2, where the hydrogels equilibrated at 20% more swelling than the initial hydrogel after 24 h (Figure 5b). Swelling occurred within the first 4 h and then remained stable until equilibrium 24 h later (Appendix A). This similarity between the two compositions of hydrogels may be due to the balanced crosslinking density in the hydrogel between the decreased molecular weight HA-Mal at twice the concentration as the increased molecular weight HA-Mal. 

Despite the rheological results indicating a relatively weak gel, the barrier assay provided information about the ability for this hydrogel to serve as a barrier, a desirable trait for many regenerative applications. In this assay, hydrogels were formed in hanging inserts with a solution of FITC-dextran in the insert above the hydrogel. After 2 h, the migration % of the FITC-dextran solution that had diffused through the hydrogel into the well was calculated by the ratio between the fluorescence units of the FITC-dextran that passed through the hydrogel and the FITC-dextran passed through inserts without hydogel. We saw an average of 14.7 ± 9.4, 5.3 ± 2.7, and 1.2 ± 0.7% migration for hydrogels HMGM 1, HMGM 1 pH 7, and HMGM 2 (Appendix A). It was interesting to see reduced diffusion rates when the hydrogel component solutions were adjusted to a pH 7. This suggests that the crosslinking reaction a subsequent crosslinking density is pH dependent and offers opportunities to manipulate crosslinking kinetics or mechanical properties based on pH alone. Compared to the HMGM 1 and HMGM 2 hydrogels, we see a significant difference (*p* = 0.03) between the passage of the FITC-dextran through the gels. This may be due to the higher molecular weighted HA-Mal forming a denser network that would prevent the passing of the FITC-dextran as easily as the HMGM 1, even though the swelling ratios may be the same, which is a direct correlation with the crosslinking density. 

Because among the applications for hydrogels are encapsulation and extended release of drugs, cytokines, or other proteins, protein release was assessed. A cumulative protein release curve over 12 days was created by quantifying the release of BSA protein (10 mg/mL) from the HMGM 1 hydrogel (Figure 5c). Protein quantification was performed using the BCA (bicinchoninic acid) assay after collecting, freezing, and storing all samples during each time point. We observed that the majority of the protein release occurred within the first week, with minimal continued releases until day 12. However, at day 12, there is still an upward trend, suggesting that protein release may be sustained for additional time if necessary. Despite the concentration of BSA added into the hydrogel was 10 mg/mL, our results show that protein quantification exceeded above 10 mg/mL. This may be due to the possible diffusion of uncrosslinked gelatin at the beginning or degradation over this period of time resulting in free soluble gelatin, being included in the assay. Since the BSA assay detects total protein, it is possible for this reason to have exceeded that amount. However, this hydrogel provides the ability to slowly release protein out of the system, about 20% per day. 

### 2.5. Cell Biocompatibility Studies

Basic cell biocompatibility was assessed by culture normal human dermal fibroblasts (NHDF) or normal human epidermal keratinocytes (NHEK) in the presence of HMGM 1 hydrogel and measuring mitochondrial metabolism over time. We recognize that this is a simple assay. In future studies, more in-depth biocompatibility assessment—in vitro and in vivo—will be performed appropriate for the given hydrogel application. Cell proliferation in the presence of the HMGM 1 hydrogel was measured using the MTS cell metabolism assay after 24, 48, and 72 h of exposure. Both keratinocytes and fibroblasts were unaffected by the presence of the hydrogel over the course of 3 days (Figure 6a). Without differences observed at any time point, quantification of cell metabolism indicated a steady increase in absorbance values over time, which are proportional to cell number. This indicated continued cell proliferation, thus suggesting a cell friendly environment, free from any toxic unreacted compounds or byproducts. To corroborate these results, phase microscopy images were obtained, which are displayed in Figure 6b and provide a visual representation of the proliferation of both the keratinocyte and fibroblast cell cultures at 0, 24, 48, 72 h with and without the HMGM 1 exposure. From these images, we can observe the corresponding cell number increases compared to the control, indicating the biocompatibility of the hydrogel materials with the cells. 

Interestingly, in early studies we observed that the hydrogel prepared with 10× PBS prevented proper growth of both of the cells, falsely suggesting that the hydrogel impeded cell proliferation (data not shown). We investigated the problem of the lack of growth by separating the components of the hydrogel, which pointed to the 10× PBS that was perturbing the overall osmolality of the media (from 354 to 549 mOsm/kg). As primary cells are very sensitive to the media salt concentration, we decided to adjust the pH of the individual components of the hydrogel to a pH 5 and 4 for HA-Mal and Gel-Mal, respectively, in order to form a crosslinked hydrogel using only 1× PBS. The resulting pH of this hydrogel maintained near neutral values (pH = 6.6–6.8), similar to the pH of the hydrogel made with 10× PBS. This improved the cell metabolism. 

## 3. Conclusions

In this study, we share the initial development and characterization of a hydrogel system comprised of maleimide-functionalized hyaluronic acid and gelatin that utilizes a bifunctional thiolated PEG crosslinker. We developed this particular formulation based on the practical needs of several translational and clinical future applications requiring (1) rapid crosslinking, (2) a cell-friendly reaction environment, (3) UV light-free polymerization, and (4) natural ECM-derived based components. In particular, we are now in the midst of deploying this hydrogel system in a number of areas, including recently initiated in vivo wound healing studies and biofabrication of tissue constructs for in vitro drug screening and disease modeling. As we progress with these applications, we expect additional optimization of the hydrogel system. For example, as described above, pH plays an important role in dictating both crosslinking readiness and kinetics, as well as cell biocompatibility. Buffering the system appropriately could simplify our current approach to modulating pH the system prior to use. Additionally, as we note above, the current hydrogel formulations were quite low in terms of mechanical properties. While we do not believe this to be a limiting factor in our immediate applications, elastic modulus plays an important role in disease progression (for example in cancer, as we have demonstrated previously [51,55]) and directly influences compatibility with biofabrication technologies such as bioprinting [22,41,42,55]. As such, further exploration of methods to manipulate the mechanical properties of these hydrogels will likely be undertaken. In the studies described herein, we lay down the foundational demonstration of material synthesis, formulation, and characterization. Importantly, our resulting hydrogel system successfully meets the set of requirements driving our efforts, by supporting nearly instantaneous thiol-maleimide-based crosslinking at a neutral pH and physiological temperature environment using HA and gelatin base components. The data presented describe a well-defined ECM-derived hydrogel system now ready for subsequent translational experimentation. 

## 4. Materials and Methods

### 4.1. Materials

Sodium hyaluronate (~100 and ~200 kDa molecular weight) was purchased from Lifecore Biomedical (Chaska, MN, USA). 1-ethyl-3-(3-dimethylaminopropyl) carbodiimide (EDC) hydrochloric salt was purchased from Thermo Fisher Scientific (Rockford, IL, USA). Gelatin from porcine skin Type A (Bloom strength 175 g), 2-Morpholinoethanesulfonic acid (MES) hydrate, *N*-Hydroxysuccinimide (NHS), 1-(2-Aminoethyl) maleimide hydrochloride salt, 1-(2-Aminoethyl) maleimide trifluoroacetate (TFA) salt, pepsin from porcine gastric mucosa (>250 units/mg, Lot #SLBP2152U), and fluorescein isothiocyanate-dextran (3–5 kDa) were obtained from Sigma Aldrich (St. Louis, MO, USA). Di-thiol PEG crosslinker (MW 3400 g/mol) was purchased from Creative PEGWorks (Durham, NC, USA).

### 4.2. Synthesis of Maleimide-Hyaluronic Acid (HA-Mal) Conjugates

Sodium hyaluronate (1.0 g, 2.48 mmol, 1.0 eq) was dissolved in 0.1 M MES Buffer (100 mL, pH 4.5) in a 250 mL single neck round bottom flask with a magnetic stir bar. EDC (0.57 g, 2.97 mmol, 1.2 eq) and NHS (0.17 g, 2.97 mmol, 1.2 eq) were first dissolved together in 10 mL of MES buffer (0.1 M, pH 4.5) and then quickly poured into the hyaluronic acid (HA) solution. The reaction mixture was stirred at 300 rpm for 30 min at room temperature (23 °C). 1-(2-Aminoethyl) maleimide HCl (0.53 g, 2.97 mmol, 1.2 eq) was dissolved in 5 mL of distilled water (for a ~10% *w/v* solution) and then added dropwise to the reaction. Following the small molecule addition, the flask was capped and then stirred at 300 rpm overnight (18–24 h). The crude mixture was dialyzed (MWCO 12–14k) in a 5 L container with acidified water (pH 3.0–3.5 using 1 M HCl) stirring slowly at 50–100 rpm for 3 days. The purified solution was adjusted to a pH of 3.0 using 1 M HCl or 1 M NaOH, and then sterile filtered through a 0.2 µM filter to remove insoluble impurities. The filtrate was set to freeze at −20 °C overnight and lyophilized (<100 mTorr, −50 °C) for 3 days or until fully dried. The HA-Mal materials were stored in the −20 °C freezer until used.

### 4.3. Synthesis of Maleimide-Gelatin (Gel-Mal) Conjugates

Gelatin (1.0 g) was dissolved in 0.1 M MES Buffer (100 mL, pH 4.5) in a 250 mL single neck round bottom flask with a magnetic stir bar (1.5–2 inches) placed in a water bath heated at 37 °C. EDC (0.575 g, 3 mmol) and NHS (0.345 g, 3 mmol) were first combined and dissolved in 10 mL of MES buffer (0.1 M, pH 4.5) and then quickly poured into the gelatin solution. The reaction mixture was stirred for 30 min at 300 rpm. 1-(2-Aminoethyl) maleimide TFA salt (1.524 g, 6 mmol) was dissolved in 10 mL of distilled water and then added dropwise to the reaction flask. Following the small molecule addition, the flask was capped and then stirred at 300 rpm overnight (18–24 h) at 37 °C. The crude mixture was dialyzed (MWCO 12–14 k) in a 5 L container with acidified water (pH 3.0–3.5 using 1 M HCl) stirring slowly at 50–100 rpm for 3 days. The purified solution was adjusted to a pH of 3.0 using 1 M HCl or NaOH, and then sterile filtered through a 0.2 µM filter to remove insoluble impurities. The filtrate was set to freeze at −20 °C overnight and lyophilized (<100 mTorr, −50 °C) for 3 days or until fully dried. The Gel-Mal materials were stored in the −20 °C freezer until used.

### 4.4. HA-Mal and Gel-Mal Chemical Characterization

#### 4.4.1. ^1^H NMR Characterization of HA-Mal and Gel-Mal

Purity and degree of substitution (DS%) was determined using ^1^H NMR spectroscopy (Bruker Avance 400 MHz spectrometer). Dried HA-Mal and Gel-Mal materials were dissolved in deuterium oxide (D_2_O, 99.9%, Cambridge Isotope Laboratories, Inc., Tewksbury, MA, USA) at 10 mg/mL. The degree of substitution (DS%) was only determined for HA-Mal, where all peak integrations were normalized to the peak at 2.0 ppm, which corresponds to the *N*-acetyl group on HA. The DS% was calculated as the ratio between the integral at 6.8 ppm divided by 2 and the sum of the integral at 2.0 ppm divided by 3 and the integral at 6.8 ppm divided by 2. Values of 2 and 3 used to divide the peak integrations represent the protons on the maleimide and the *N*-acetyl groups, respectively. 

#### 4.4.2. UV–Vis Spectroscopy Characterization of HA-Mal and Gel-Mal

UV–Vis spectroscopy was used to display the maleimide group addition on HA and gelatin as well as to determine the maleimide concentration in both HA-Mal and Gel-Mal. Materials were dissolved in DI water at a concentration of 1 mg/mL and solutions were measured in plastic cuvettes using the UV Spec 2600 Spectrophotometer (Shimadzu Scientific Instruments Inc., Kyoto, Japan). Absorbance values were first measured between 200–400 nm to determine the maximum absorbance values of the maleimide group. Then, solutions of varying known concentrations of maleimide were used to create a standard curve to determine the maleimide concentration (µmol/mg) in HA-Mal and Gel-Mal. 

The stability of the maleimide group was investigated in 0.1 M MES buffer at various pH conditions (pH 3–8). MES buffer solutions were adjusted at various pH conditions with NaOH and HCl and then used to dissolve maleimide (99%, Sigma Aldrich, St. Louis, MO, USA) at 10 mg/mL (*n* = 3). Then, solutions were diluted (1:50) and absorbance (λ_max_ = 295 nm) was measured at 1, 24, 48, 72, and 96 h. It should be noted that the pH of each maleimide solution was monitored during this time study. Absorbance values were compared across pH conditions to demonstrate maleimide sensitivity to pH.

#### 4.4.3. Molecular Weight Quantification of HA-Mal with SEC-MALS

Unmodified and maleimide-modified HA samples were dissolved in 1× PBS, centrifuged at 20,000 RCF for 10 min, and filtered through a 0.45 µm nylon milter unit. Samples were injected into an HPLC system composed of a Waters 717 plus autosampler, two Waters model 510 chromatography pumps, and three detectors in series (MALS detector at 661 nm, differential refractive index detector at 658 nm, UV/Vis detector monitoring at 214 nm). Data acquisition and analysis were performed using Wyatt ASTRA software to determine the average molecular weight of the polymer.

### 4.5. HA and Gelatin Hydrogel Formulation

Hydrogels (HMGM 1, 2) for all characterization studies were formed using materials dissolved in 10× PBS, except for the cell proliferation assay. The crosslinked hydrogel (HMGM 1) tested in the cell proliferation assay used 1× PBS to dissolve HA-Mal and Gel-Mal, which were adjusted to pH 5 and pH 4, respectively, post-dialysis. HA-Mal, Gel-Mal, and PEG-dithiol (PEGDSH) crosslinker were initially dissolved at 1.0%, 1.6%, and 0.5% *w*/*v*, respectively and combined at a 2:1:1 ratio. Due to the rapid crosslinking reaction, HA-Mal and Gel-Mal solutions were first combined and then crosslinked with PEGDSH.

### 4.6. Hydrogel Gelation Testing

#### 4.6.1. Visual Gelation Confirmation by Vial Inversion

Hydrogel precursor solutions were prepared without a crosslinker, the HMGM 1 formulation, and the HMGM 2 formulation, in 5-dram glass vials and allowed time sufficient for crosslinking to occur (in crosslinkable groups). Vials were then inverted to allow uncrosslinked solutions to flow. In comparison, crosslinked gels hold their shape at the end of the vial. 

#### 4.6.2. Nuclear Magnetic Resonance Evaluation

NMR spectroscopy (Avance II 400 MHz spectrometer, Bruker Corporation, Billerica, MA, USA) was used to confirm the thiol-maleimide reaction by comparing proton spectra of pre- and post-crosslinked solutions and observing the presence of the maleimide peak at 6.8 ppm All materials were dissolved in D_2_O at the corresponding concentrations and part ratios as described in Section 4.5. Resulting solutions were combined in the same NMR tube, vortexed for 10 s, and allowed to react for 5 min prior to reading samples in the instrument. 

### 4.7. Hydrogel Physical Characterization

#### 4.7.1. Rheological Mechanical Characterization

Rheological mechanical properties of the hydrogel (HMGM 1) were examined with an oscillatory strain test using a Discovery HR-2 rheometer (TA Instruments, New Castle, DE, USA) and an 8 mm parallel plate geometry, roughened with sandpaper, similar to previous studies by our team that characterized HA-based hydrogels [39,40]. Measurements were conducted at 23 °C; a gap height of 1200 µm; an axial force of 0.05 ± 0.01 N. Hydrogels (80 µL, *n* = 3) were prepared as described in Section 4.5 in 8 mm PDMS molds and allowed to completely crosslink for 5 min prior to transferring hydrogel disc onto the roughed platform. The amplitude strain sweep was performed with an oscillating strain range of 0.1–100% and an oscillating frequency of 1 Hz.

#### 4.7.2. Hydrogel Swelling

Aliquots of 500 µL of each hydrogel formulation were allowed to crosslink for 10 min at the bottom of replicate pre-weighed glass vials (*n* = 5). Initial weights of vials with hydrogels were measured (initial mass). Then, 1 mL of PBS was added on top of each gel and incubated at 37 °C. Swelling in this set up occurred uniaxially vertically. At 1, 2, 3, 6, 24 h, the buffer solution from each vial was carefully removed, the vials were weighed (swollen mass), and then fresh PBS was replaced. The masses of the original and swollen hydrogels were calculated by subtracting the mass of the empty vials from the total masses. The mass swelling ratio (%) was then calculated as the ratio of the difference between the swollen and initial masses to the initial mass. 

#### 4.7.3. Cumulative Total Protein Release

Hydrogels (*n* = 5) as described in Section 4.5 were prepared with BSA protein (final concentration of 10 mg/mL) in 0.5 mL volumes in a 24-well plate. PBS (0.5 mL) was added to the top of each hydrogel and the plate was incubated at 37 °C. At 24-h increments, the buffer solutions were frozen for storage at −20 °C and then replaced with fresh PBS for 12 days. The collected samples were used to quantify the total protein content using a Pierce BCA Protein Assay Kit (Thermo Fisher Scientific, Rockford, IL, USA) and the resulting data was used to generate a cumulative protein release curve.

#### 4.7.4. Hydrogel Barrier Function Assay

Hydrogels (100 µL, *n* = 5) were prepared in Millicell 3.0 µM PET hanging cell culture inserts (MilliporeSigma, Burlington, MA, USA) in a 24-well plate. A solution of FITC-dextran (MW 3–5 kDa) was made in Dulbecco’s Phosphate Buffered Saline (DPBS) (5 mg/10 mL) and gently layered on top of the crosslinked hydrogel in each insert. Hanging inserts filled with only FITC-dextran was the control. Once DPBS (900 µL) was placed in the receiving wells of the 24 well plate, the plate was covered in foil and incubated at RT for 2 h. The hanging inserts were removed and the solution in the receiving well were mixed before transferring 200 µL of solution to a black, flat bottom 96 well plate. Fluorescence was measured at λ_ex_ = 490/λ_em_ = 520 nm and the barrier function of the hydrogel was determined by the percent migration of the FITC-Dextran molecule through the hydrogel. This was described as the ratio of the relative fluorescence units (RFU) of the hydrogel samples and the RFU values of the FITC-dextran only control. 

### 4.8. Human Dermal Fibroblasts and Epidermal Keratinocytes Cell Culture

Primary normal human dermal fibroblasts were obtained from a single adult donor (PromoCell, Lot #458Z020.2) and cultured in Fibroblast growth medium 2 (PromoCell GmbH, Heidelberg, Germany) supplemented with fetal calf serum (0.02 mL/mL), recombinant human basic fibroblast growth factor (1 ng/mL), and recombinant human insulin (5 µg/mL). When fibroblasts reached 70–80% confluency (37 °C, 5% CO_2_), they were detached with TrypLE Express Enzyme (1X) without phenol Red (Gibco, Thermo Fisher Scientific, Rockford, IL, USA) for 3–5 min, centrifuged at 200× *g*, and counted for the cell proliferation assay (passage 3). 

Normal human epidermal keratinocytes were pooled from 3 adult donors (PromoCell GmbH, Heidelberg, Germany, Lot #418Z026) and cultured in keratinocyte serum-free growth medium 2 (PromoCell GmbH, Heidelberg, Germany) with supplements of bovine pituitary extract (4 µL/mL), recombinant human epidermal growth factor (0.125 ng/mL), recombinant human insulin (5 µg/mL), hydrocortisone (0.33 µg/mL), epinephrine (0.39 µg/mL), recombinant human transferrin (10 µg/mL), and calcium chloride (0.06 mM). When keratinocytes reached 60–75% confluency (37 °C, 5% CO_2_), they detached with Hyclone 0.05% Trypsin protease solution with EDTA and phenol red (Cytiva, Marlborough, MA, USA) for 5–7 min, neutralized with 5% FBS, centrifuged at 150× *g*, and counted for the cell proliferation assay (passage 3).

### 4.9. Cell Proliferation Assay

Fibroblasts and keratinocytes were prepared as above and seeded separately into 24-well plates at a density of 6.5k/cm^2^. After an overnight incubation to allow cell attachment, Millicell 3.0 µM PET hanging cell culture inserts (MilliporeSigma, Burlington, MA, USA) were transferred into the corresponding wells. Control groups (*n* = 3 per timepoint) included cells and hanging inserts with 1× DPBS (100 µL) and the respective cell medium (100 µL); experimental groups (*n* = 3 per timepoint) included cells and hanging inserts with hydrogels (HMGM 1) prepared as described in Section 4.5 (100 µL) with the respective medium (100 µL) on top. Metabolism was quantified using the 3-(4,5-dimethylthiazol-2-yl)-5-(3-carboxymethoxyphenyl)-2-(4-sulfophenyl)-2*H*-tetrazolium (MTS) assay (Promega, Madison, WI, USA) at 24, 48, and 72 h after sample exposure. Each well was replaced with MTS working solution (20% *v*/*v* with media, 300 µL) and the plate was incubated (37 °C, 5% CO_2_) for 1 h. Aliquots (100 µL) were transferred into a white, clear flat bottom 96 well plate and absorbance values were measured at 490 nm on a plate reader (SpectraMax i3x MiniMax 300, Molecular Devices, San Jose, CA, USA). It should be noted that media was not changed during the time course of the MTS assay. In addition, phase constrast photographs were captured of the cells within wells to visually assess confluency using a Zeiss Axiovert invertedd microscope.

## Figures and Tables

**Figure 1 gels-07-00013-f001:**
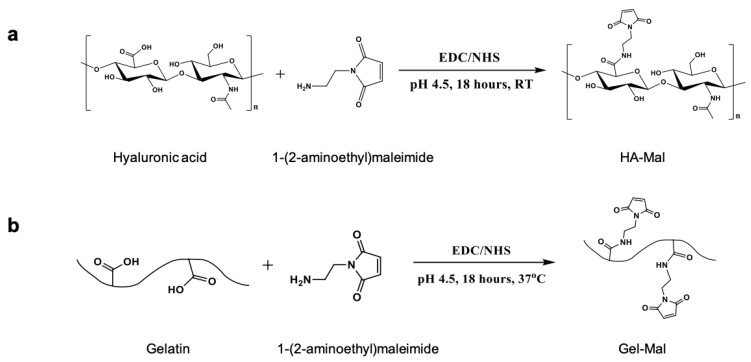
Reaction schemes of the EDC/NHS activated syntheses of (**a**) HA-Mal and (**b**) Gel-Mal.

**Figure 2 gels-07-00013-f002:**
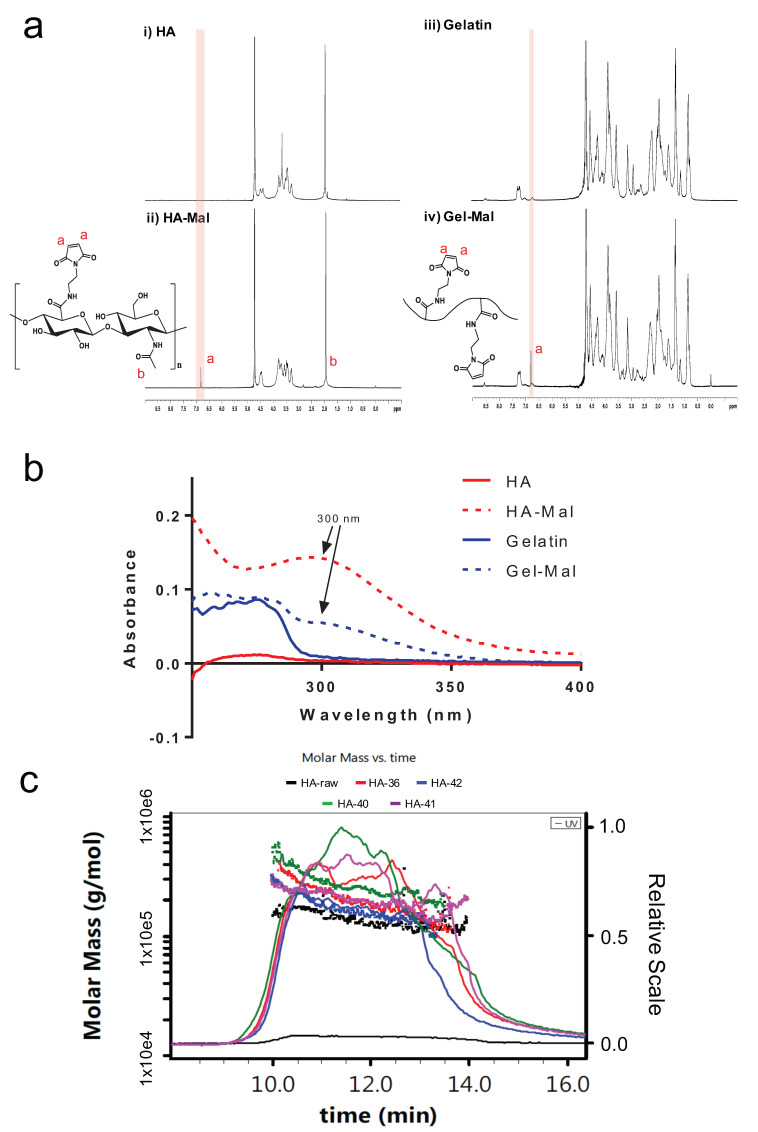
Chemical characterization of HA-Mal and Gel-Mal. (**a**) ^1^H NMR spectrum of unmodified HA and HA-Mal (left) and unmodified gelatin and Gel-Mal (right). The peak corresponding to the maleimide hydrogens (red ‘a’) is at 6.8 ppm. (**b**) UV–Vis absorption spectrum (200–400 nm) for HA, HA-Mal, Gelatin and Gel-Mal. Maximum absorbance values of the maleimide group in both HA-Mal and Gel-Mal occurred at a wavelength of 300 nm (black arrow). (**c**) SEC-MALS spectrum of HA and HA-Mal for MW determination.

**Figure 3 gels-07-00013-f003:**
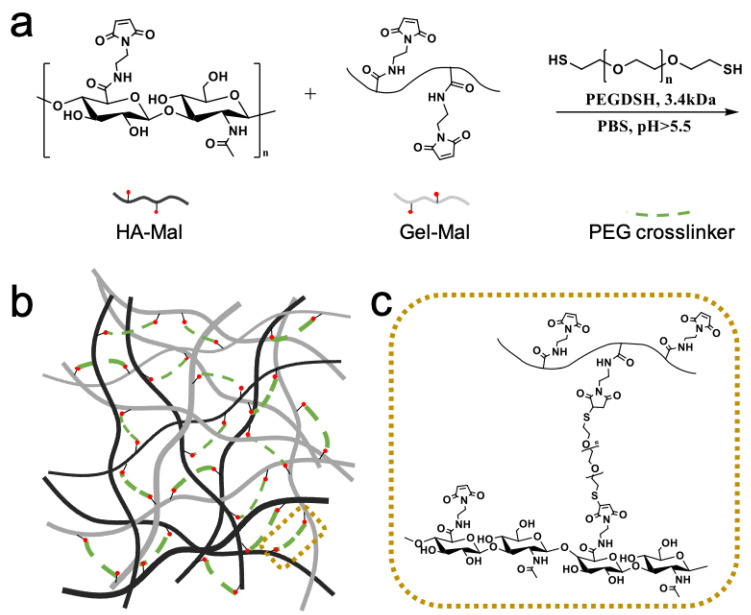
Hydrogel formulation. (**a**–**c**) Schematic representation of the rapid hydrogel crosslinking reaction with HA-Mal, Gel-Mal, and PEGDSH in PBS. Maleimide groups (red circle) on both HA (black line) and gelatin (gray line) undergo Michael addition reactions with thiols on the PEG crosslinker (dotted green line), resulting in a crosslinked macromolecular network (dotted gold outline).

**Figure 4 gels-07-00013-f004:**
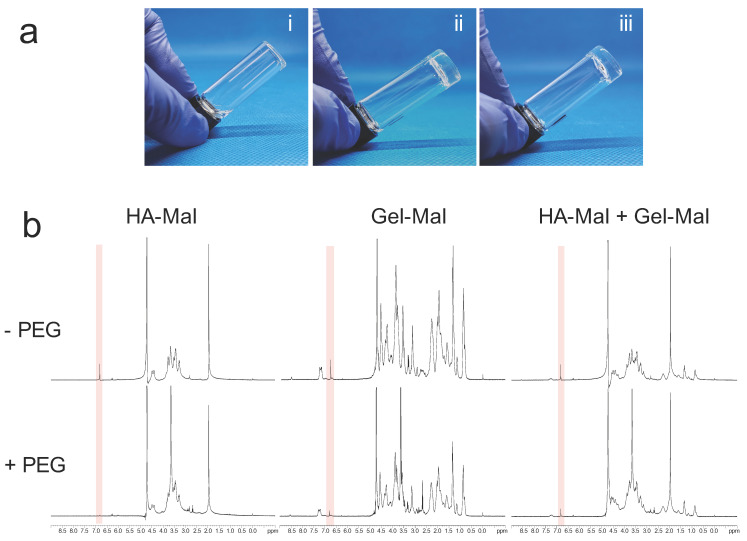
Gelation Testing with HA-Mal, Gel-Mal, and PEGDSH. (**a**) Visual gel formation of hydrogel (i) without PEGDSH crosslinker (ii) HMGM 1 (iii) HMGM 2. Materials crosslinked in <5 s. (**b**) ^1^H NMR spectrum of HA-Mal with PEGDSH (left), Gel-Mal + PEGDSH (middle), and HA-Mal + Gel-Mal + PEGDSH (right). The maleimide peak is reduced or eliminated at 6.8 ppm (pink highlighted regions) with the addition of the PEG crosslinker.

**Figure 5 gels-07-00013-f005:**
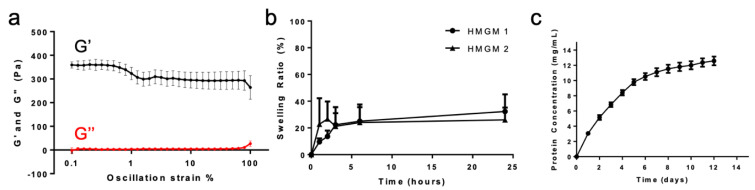
Physical HMGM Hydrogel Characterization. (**a**) Oscillation strain (0.1–100%) rheological results. Storage modulus (G′) and loss modulus (G″) are indicated. (**b**) Swelling Ratio %. (**c**) Cumulative total protein release over 12 days.

**Figure 6 gels-07-00013-f006:**
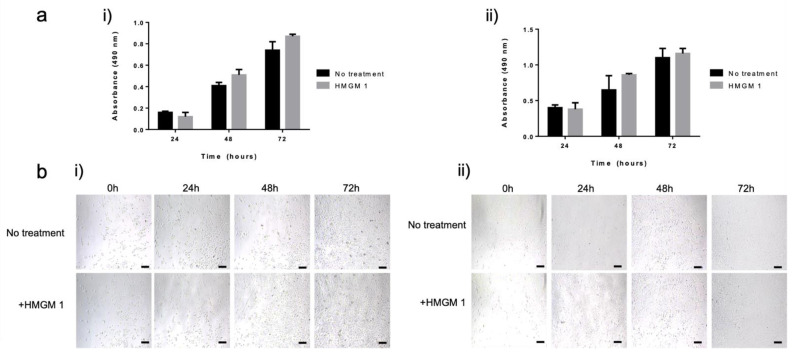
Proliferation of NHEK and NHDF cells over time under normal culture conditions and exposed to HMGM 1 hydrogels. (**a**) Relative proliferation of (i) NHEK and (ii) NHDF cells. Absorbance is proportional to cell number, quantified by measuring mitochondrial metabolism (MTS assay) at 24, 48, and 72 h. (**b**) Phase microscopy images showing plated (i) NHEK and (ii) NHDF at 0, 24, 48, and 72 h. Scale bar—50 µm.

## Data Availability

The authors confirm that the data supporting the findings of this study are available within the article and its Appendix A.

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
