# Peer review of "A Rapid Crosslinkable Maleimide-Modified Hyaluronic Acid and Gelatin Hydrogel Delivery System for Regenerative Applications"

_gels, 2021, doi:10.3390/gels7010013_

Round 1

Reviewer 1 Report

Yoo and coworkers developed a rapid crosslinkable hydrogel delivery system for regenerative application. The idea and motivation are reasonable, and the characterizations are well performed. Therefore, I suggest it can be published in Gels, subject to some revision as below.

  1. The polydispersity of HA before and after modification should be added.
  2. While the degree of substitution (DS%) of HA-Mal confirmed by NMR is 7.2%, what’s the DS% of Gel-Mal?
  3. Why the molecular weight of PEG3400 is chosen in this study?
  4. The image resolution of figure 2 is not high enough. The authors should make efforts to improve it.
  5. Several recent studies (Acta Biomaterialia 2020, 101, 1-13; Science 2020, 370 (6514), 335-338; Biomacromolecules 2020, 21, 4663–4672) should be included in the application of the hydrogels (line 40-41).
  6. Format issue. For example: ‘Hydrogels’ (line 15).

Author Response

Yoo and coworkers developed a rapid crosslinkable hydrogel delivery system for regenerative application. The idea and motivation are reasonable, and the characterizations are well performed. Therefore, I suggest it can be published in Gels, subject to some revision as below.

  1. The polydispersity of HA before and after modification should be added.

Polydispersity, while useful in terms of specific polymer chemistry – usually more in synthetic polymers – is a feature that our team, who have worked with hyaluronic acid biomaterials for over 15 years, has never sought to define, simply due to the fact that natural polymers before and after modification have a large range in MW. We would argue that the polydispersity of a relatively large MW polymer/polysaccharide in a crosslinked hydrogel network is not as informative as for synthetic polymers, as the polymer/polysaccharide chains once crosslinked form a macromolecular network in which polydispersity is no longer has a mjor impact on the macro properties.

  1. While the degree of substitution (DS%) of HA-Mal confirmed by NMR is 7.2%, what’s the DS% of Gel-Mal?

DS% in gelatin products is inherently more complicated than in HA (or other polysaccharide or synthetic polymer) products. This is due to the heterogeneity of functional groups in gelatin (and collagen) based on a repeating random amino acid sequence. However, we have in previous studies relied on the fact that gelatin can be modified using similar strategies to HA that target carboxylic acid groups for other modifications. This has been a tried-and-true method that has yielded commercially available hydrogel systems that are widely used. Furthermore, our gelation testing and material property characterization indicate sufficient DS% to form free standing, yet pliable gels that are user friendly.  

  1. Why the molecular weight of PEG3400 is chosen in this study?

We used a PEG crosslinker of this particular molecular weight based on our experience with other PEG crosslinkers (PEGDA specifically) in other hydrogel systems. The 3400 MW crosslinker supports terminal hydrogel elastic moduli of 200 to 400 Pa – about 300 Pa on average in this study. We have found that this hydrogel elastic moduli range supports results in a hydrogel that is soft, pliable, and if used in a wound healing/regeneration setting can “move” with the biological tissue. We have demonstrated this utility in a number of previous publications, which we cited in our manuscript. (doi: 10.1002/jbm.b.33736.; doi: 10.1002/sctm.17-0053.; doi: 10.1002/sctm.19-0101.) However, we have added text in our revision to clarify this rationale. In addition, Supplemental Table 2 describes testing of other MW and geometry crosslinkers to confirm our choice of using the 3400 MW crosslinker.

  1. The image resolution of figure 2 is not high enough. The authors should make efforts to improve it.

We have reworked Figure 2 in order ensure readability is sufficient.

  1. Several recent studies (Acta Biomaterialia 2020, 101, 1-13; Science 2020, 370 (6514), 335-338; Biomacromolecules 2020, 21, 4663–4672) should be included in the application of the hydrogels (line 40-41).

Thank you for these suggestions. They have been integrated into our revision.

  1. Format issue. For example: ‘Hydrogels’ (line 15).

Thank you for bringing this to our attention. This has been corrected.

Reviewer 2 Report

Review comments for gels-1083876 entitled “A Rapid Crosslinkable Maleimide-modified Hyaluronic Acid and Gelatin Hydrogel Delivery System for Regenerative Applications” described by Yoo KM et al.

The authors created maleimide-modified hyaluronic acid and gelatin hydrogel system using bifunctional thiolated PEG crosslinker. The hydrogel, which is natural ECM-based materials, include some advantages such as rapid UV light-free crosslinking kinetics, and cytocompatibility. In this study, maleimide-modified polymers evaluation, gelation, gel characterization, and biocompatibility were evaluated. From the text, the reader immediately assumes the UV free gelation is useful for clinical application.

Gelation of the system possibly depends on molecular weight of PEGdSH. The authors described that “then combining them at varying ratios with thiolated PEG crosslinkers of several weights (1k, 3.4k, 10k) and arms (2, 4, 8), 2 formulations were developed that resulted in a self-standing, easily controllable hydrogel (Supplementary Table 1)” in lane 199 to 201. However, there were only 3.4 k in supplementary table 1. The author should add data about other weights of PEGdSH.

In fig 6b, the photographs are unclear. The authors should stain the cells with dye and replaced more magnified one to recognize cells and contaminants. And also, materials and methods section is lack the explanation of this experimental condition. Moreover, there are no scale bars in fig 6b.

Author Response

  1. The authors created maleimide-modified hyaluronic acid and gelatin hydrogel system using bifunctional thiolated PEG crosslinker. The hydrogel, which is natural ECM-based materials, include some advantages such as rapid UV light-free crosslinking kinetics, and cytocompatibility. In this study, maleimide-modified polymers evaluation, gelation, gel characterization, and biocompatibility were evaluated. From the text, the reader immediately assumes the UV free gelation is useful for clinical application.

We reworded the section we believe the reviewer is referring to (end of the Introduction) to try to clarify this point. UV photopolymerization has great gelation kinetics and usability characteristics. However, clinicians we have worked with desired a UV-free crosslinking method to avoid the requirement to have portable UV light-sources in the operation room or doctors office, as well as a desire to minimize patient exposure to UV irradiation.

  1. Gelation of the system possibly depends on molecular weight of PEGdSH. The authors described that “then combining them at varying ratios with thiolated PEG crosslinkers of several weights (1k, 3.4k, 10k) and arms (2, 4), 2 formulations were developed that resulted in a self-standing, easily controllable hydrogel (Supplementary Table 1)” in lane 199 to 201. However, there were only 3.4 k in supplementary table 1. The author should add data about other weights of PEGdSH.

We have added a Supplementary Table (now Supplementary Table 2) including these different conditions and qualitative assessments of the resulting hydrogels. 

  1. In fig 6b, the photographs are unclear. The authors should stain the cells with dye and replaced more magnified one to recognize cells and contaminants. And also, materials and methods section is lack the explanation of this experimental condition. Moreover, there are no scale bars in fig 6b.

We have updated Figure 6b to better visualize the cells. Scale bars were also added.

In terms of the materials and methods, these images were simply taken in parallel to the MTS proliferation data. This is described briefly at the end of that particular section. However, we have included additional information such as the microscope and imaging type.